

# Effects of drought and salt stress on seed germination and seedling growth of *Elymus nutans*

Jianting Long, Mengjie Dong, Chuanqi Wang and Yanjun Miao

Tibet Agricultural and Animal Husbandry University, Tibet, China

## ABSTRACT

Drought and soil salinization are global environmental issues, and *Elymus nutans* play an important role in vegetation restoration in arid and saline environments due to their excellent stress resistance. In the process of vegetation restoration, the stage from germination to seedling growth of forage is crucial. This experiment studied the effects of PEG-6000 simulated drought stress and NaCl simulated salinization stress on the germination of *E. nutans* seeds, and explored the growth of forage seedlings from sowing to 28 days under drought and salinization stress conditions. The results showed that under the same environmental water potential, there were significant differences in responses of seed germination, seedling growth, organic carbon, total nitrogen and total phosphorus of above-ground and underground parts of *E. nutans* to drought stress and salinization stress. Using the membership function method to comprehensively evaluate the seed germination and seedling indicators of *E. nutans*, it was found that under the same environmental water potential, *E. nutans* was more severely affected by drought stress during both the seed germination and seedling growth stages. *E. nutans* showed better salt tolerance than drought resistance.

Subjects Agricultural Science, Plant Science, Environmental Contamination and Remediation
Keywords *Elymus nutans*, Drought stress, Salt stress, Seedling growth

## INTRODUCTION

At present, soil salinization caused by drought, unreasonable irrigation and other factors is a global environmental problem that needs to be solved urgently, and soil drought and salinization are the main abiotic stress factors affecting crop growth (*Dai, 2013*; *Pokhrel et al., 2021*). The arid zone in China accounts for 1/3 of the total area, including a large amount of saline soil. The important factors affecting agricultural development and ecological environment protection in arid zone are drought, soil salinization, and secondary salinization caused by irrigation (*Zhang et al., 2022*; *Zhao et al., 2022*).

The unique and complex and diverse climate of Tibet, with high altitude, low temperature, large diurnal temperature difference; low precipitation and large regional differences; strong sunlight and long daylight hours, is the main reason for its susceptibility to drought and soil salinization, and the annual crop yield reduction caused by drought and water shortage or soil salinization limits the development of agriculture in the region (*Zhang et al., 2021*; *Javed et al., 2021*; *Kong et al., 2018*).

Corresponding author
Yanjun Miao, myj666@126.com

*Elymus nutans* is a perennial grass of the family Gramineae, distributed in Tibet, Hebei, Qinghai, Sichuan, Shaanxi, Gansu, Xinjiang, and Inner Mongolia in China, and also in Turkey, Mongolia, India, and Russia (*Liu, Tao & Dou, 2021*; *Ma et al., 2006*). *E. nutans* has strong growth ability, high crude protein and fat content, and good palatability. And it plays an important role in the improvement of alpine grasslands and the construction of artificial grasslands (*An et al., 2022*; *Sun et al., 2022*).

Through long-term natural selection, *E. nutans* has gradually developed resistance to cold, drought, salinity and other high quality genetic characteristics, which not only have high forage value, but also can play an important role in resisting wind and sand damage and conserving water and soil (*Li et al., 2021*; *Luo et al., 2019*).

Soil drought and salinization, as two major abiotic stress factors affecting plant growth and crop yield, both belong to environmental water potential stress (osmotic stress) on plants in a certain sense (*Du et al., 2022*; *Bashir et al., 2019*). Seed germination is the beginning of spermophyte life history and the most sensitive period to the environment (especially in arid and saline environments), while in some arid and semi-arid areas seed germination is subjected to both drought and salt stress (*Wijewardana et al., 2019*; *Silva et al., 2018*). The successful establishment of vegetation communities is directly determined by the success of seedling formation after seed germination, and seedlings also respond to environmental stress during their growth process (such as adjusting plant height, changing root structure, *etc.*) (*Sun, He & Li, 2019*; *Andivia et al., 2021*). Current research has shown (*Wang, Liu & Zhang, 2021*) that both drought stress and salt stress can affect the germination rate of *E. nutans* seeds, with higher concentrations of PEG-6000 and NaCl directly inhibiting the germination of *E. nutans* seeds. Moreover, PEG-6000 and NaCl stress led to the increase of proline, malondialdehyde and soluble protein content, and the increase of superoxide dismutase, peroxidase and catalase activities inhibited the growth of *E. nutans* seedlings (*Wang, Wu & Yu, 2017*; *Song, Yang & Jing, 2022*). In order to study the response of *E. nutans* seed germination and seedling formation to drought and salt stress, we conducted a pot experiment in the artificial climate incubator of the Grassland Laboratory of Tibet Agricultural and Animal Husbandry College in Linzhi, Tibet. We studied the effects of drought stress and salt stress on the germination and seedling growth of *E. nutans* seeds under the same osmotic potential, and measured the germination rate, germination potential, root length, bud length, and germination index of *E. nutans* seeds, seedling growth indicators and the content of organic carbon, total nitrogen, and total phosphorus in the aboveground and underground parts. We explored the similarities and differences in the response of *E. nutans* to drought stress and salinization stress, comprehensively evaluated the degree of drought stress and salt stress suffered by *E. nutans* seeds and seedlings, and clarified the tolerance of *E. nutans* seeds and seedlings to drought and salt stress under the same environmental water potential, in order to provide reference for vegetation restoration in arid and salinized areas.

## MATERIALS AND METHODS

### Material

The experiment was completed in the Grass Science Laboratory of Tibet Agricultural and Animal Husbandry College, Tibet Autonomous Region (29°39′57.5″N, 94°20′31.4″E). The test seeds are *Elymus nutans*. cv. Baqing, provided by the Grass Science Laboratory of Tibet Agricultural and Animal Husbandry College. The experiment commenced on October 18, 2022, and the germination experiment ended on October 28, totaling 10 days. And the seedling experiment ended on November 15th for a total of 28 days.

### Experimental design

#### Germination test

PEG-6000 solution (simulated drought stress) and NaCl solution (simulated salt stress) were prepared with water potential of 0 (CK), −0.04, −0.14, −0.29, −0.49, −0.73, −1.02 MPa, respectively (*Michel & Kaufmann, 1973*; *Michel & Radcliffe, 1995*). The experiment used the article germination method. After soaking the filter article in the stress solution mentioned above, it was placed in a Petri dish. A total of 50 seeds were evenly placed in each culture dish as one replicate, and each treatment setting was repeated three times. To ensure a constant water potential in the environment, culture dishes were placed in a 25 °C constant temperature incubator, and each dish was weighed. Distilled water was added the next day to reach a constant weight. The artificial climate incubator has been set with a light intensity of 1,250 lx, a light cycle of 12 h, and a dark cycle of 12 h. The experiment considered root length or bud length equal to seed length as germination, and the number of sprouts was counted daily and moldy seeds were promptly treated.

#### Seedling test

The artificial climate incubator and solution preparation settings are the same as 2.2.1. The experiment used sand culture as culture medium, each germination box (19 × 13 × 12 cm) was loaded with 1,000 g of dry sand, 100 ml of different gradients of PEG-6000 and NaCl solutions were added. After stirring evenly, sow at a depth of 2–3 cm and a seeding rate of 5 g·m$^{-2}$. To guarantee the water potential of the sand culture environment, the distilled water was replenished three times a day, morning, noon and night, according to the evaporation in the germination box, using the weighing method to ensure that the solution content in the sand culture was around 10%. Hoagland Total Nutrient Solution was replenished every 7 days, 20 ml each time. Shoots extending out of the sand surface were considered as emergence. After all seedlings have emerged, the height of the aboveground part was recorded. After 28 days, the root morphology indicators and the content of organic carbon, total nitrogen, and total phosphorus in the aboveground and underground parts were measured.

### Measurement indexes and methods

#### Germination test index

The germination percentage, germinating energy, root length, shoot length and other indicators related to seed germination were determined and equated as follows:

Germination percentage (G, %) = (number of normally germinated seeds within 10 d of final germination/number of seeds for testing) × 100%;

Germinating energy (GE, %) = (Number of seeds that germinated normally on the 5th day of germination test/number of seeds for testing) × 100%;

Germination index (GI) = $\sum$ (Gt/Dt), Gt means the number of seeds germinated on day t and Dt means the corresponding number of days to germination;

Vigor index (VI) = germination index × Sx, Sx is the mean shoot length;

Shoot length and root length: After the germination test, 10 seedlings were randomly selected from each treatment petri dish and the test shoot length and root length were measured using a straightedge (1 mm). Subsequent use for variance analysis.

### Seedling test index determination and methods

Ten seedlings were randomly selected from the germination boxes of each treatment, and the aboveground height of 10 seedlings was measured daily. The trend of plant height change was plotted using the average value. The plant growth rate was calculated every 7 days for a total of four cycles. The plant height growth rate was R = (L2 − L1)/7, where L2 refers to the plant height measured on the last day of each cycle and L1 refers to the plant height measured on the first day of each cycle. The total root length, surface area, volume and number of root tips were determined by EPSON GT-X980 root scanner; plant height and root length were determined by straightedge. After measuring the morphological indicators, *Elymus nutans* was dried in a 65 °C oven for 48 h, and total nitrogen content was determined by the Kjeldahl method (*Singh et al., 2020*); total phosphorus content was determined by molybdenum blue ascorbic acid method (*Li et al., 2022*); determination of organic carbon content was made by Elementar vario TOC elemental analysis (*Wolski et al., 2022*).

## Data statistics and analysis

Germinating energy, root length, shoot length, vigor index, and germination index under PEG-6000 *vs* NaCl solution stress was compared using one-way ANOVA of SPSS, respectively. SPSS was used to analyze seedling plant height, growth rate, total root length, root tip number, root surface area, root volume, leaf root organic carbon content, root organic carbon content, leaf total nitrogen content, root total nitrogen content, leaf total phosphorus content, root total phosphorus content, leaf C:N, root C:N, leaf N:P, root N:P under PEG-6000 and NaCl solution stresses, respectively. The significance was analyzed by one-way ANOVA. Principal component analysis and correlation analysis were performed using Origin Pro 2021 on the environmental water potential, seedling height, total root length, root surface area, root volume, leaf root organic carbon content, root organic carbon content, leaf total nitrogen content, root total nitrogen content, leaf total phosphorus content, and root total phosphorus content of PEG-6000 and NaCl solutions, respectively. A dose-response meta-analysis was conducted on seed germination rate using SPSSAU, based on the number of tested seeds, germination number, and environmental water potential. The drought and salt resistance of *E. nutans* seeds and seedlings were analyzed and compared using the affiliation function method. The index was calculated as:

$X(\mathrm{u}) = (X - X_{\min})/(X_{\max} - X_{\min})$, and if the index was negatively correlated with drought resistance then the formula was: $X(\mathrm{u}) = 1 - (X - X_{\min})/(X_{\max} - X_{\min})$.

## RESULTS

### Effects of treatments on seed germination

The PEG-6000 solution simulated drought stress affected the germination rate of *E. nutans*. When treated at −0.14 MPa, the germination rate of *E. nutans* seeds decreased by 15.33% compared to the control group ($P < 0.01$). With the decrease of environmental water potential, the germination rate of *E. nutans* seeds under PEG-6000 solution stress showed a significant decrease trend ($P < 0.05$). The lowest germination rate was only 26% under the treatment of −1.02 MPa (Fig. 1A). For germination potential, when the environmental water potential was lower than −0.14 MPa, the germination potential was significantly lower ($P < 0.01$) than CK, with a minimum of −1.02 MPa and a germination potential of 7.33 (Fig. 1B). In terms of shoot length, with the decrease of environmental water potential, the shoot length of *E. nutans* showed a significant decrease trend, and all treatments were significantly lower than CK ($P < 0.01$) (Fig. 1C). In terms of root length, there was no significant difference between the −0.04 and −0.14 MPa treatments and CK, while the other treatments were significantly lower than CK ($P < 0.01$) (Fig. 1D). The germination index showed a very significant downward trend with the decrease of environmental water potential ($P < 0.01$), with the lowest being −1.02 MPa treatment and the germination index being 11.67 (Fig. 1E). The vitality index and germination index showed a very significant downward trend with the decrease of environmental water potential ($P < 0.01$), and the lowest germination index was 22.24 when treated with −1.02 MPa (Fig. 1F).

The NaCl solution simulated salt stress affected the germination of *E. nutans* seeds. Among them, the −0.04 MPa treatment had a 2.77% higher germination rate than CK treatment. As the environmental water potential decreased, the germination rate also showed a decreasing trend. When the environmental water potential was less than −0.49, −0.73, and −1.02 MPa treatment, the seed germination rate was significantly lower than CK treatment ($P < 0.01$), with the lowest being −1.02 MPa treatment, and the germination rate was 41.33% (Fig. 1A). In terms of germination potential, when the environmental water potential was less than −0.29 MPa, it was significantly lower than CK ($P < 0.01$), with a minimum of −1.02 MPa and a germination potential of 11.33 (Fig. 1B). For shoot length, with the decrease of environmental water potential, the shoot length of *E. nutans* showed a significant decrease trend, and all treatments were significantly lower than CK ($P < 0.01$) (Fig. 1C). For root length, there was no significant difference between −0.04 and −0.14 MPa treatments and CK, while the other treatments were significantly lower than CK ($P < 0.01$) (Fig. 1D). The germination index decreases with the decrease of environmental water potential, and is significantly lower than CK when treated with −0.14 MPa ($P < 0.01$). The lowest germination index is 16.29 when treated with −1.02 MPa (Fig. 1E). The vitality index showed a very significant decrease with the decrease of environmental water potential ($P < 0.01$), with the lowest being −1.02 MPa treatment and the vitality index being 49.71 (Fig. 1F).

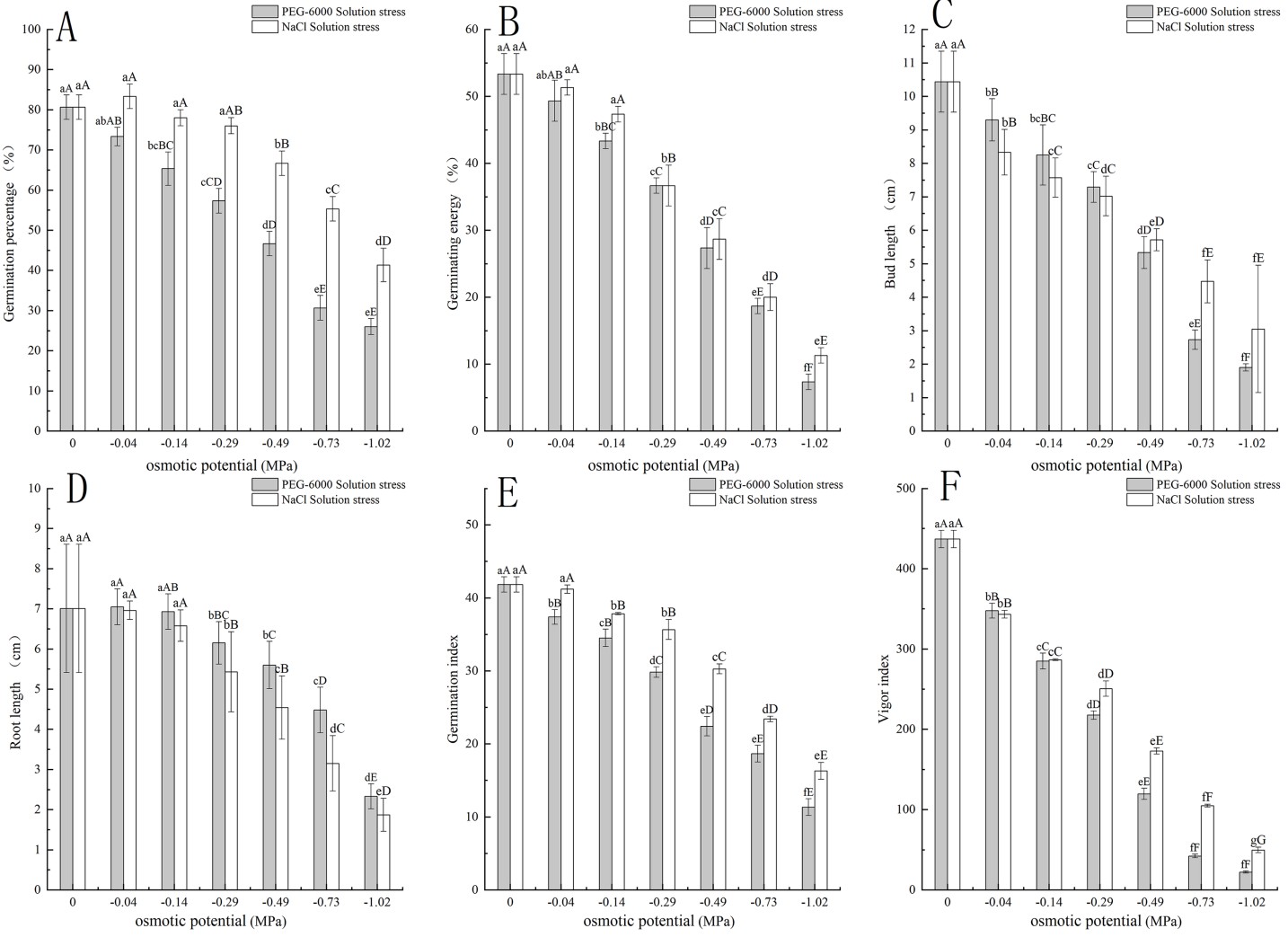

**Figure 1 Effects of two solutions on seed germination.** Two types of solution stress affect *E. nutans* seed germination rate (A), germination potential (B), bud length (C), root length (D), germination index (E), and vitality index (F). Different lowercase letters indicate significant differences in the same solution under different treatments (*P* < 0.05), different capital letters indicate that the differences in the same solution under different treatments reach a highly significant level (*P* < 0.01).

## Effects of treatments on plant height and root structure

Under simulated drought stress with PEG-6000 solution, the plant height and root structure of *E. nutans* showed a decreasing trend with the decrease of environmental water potential within 7–28 days (Fig. 2A). In terms of plant height growth rate, CK was significantly higher in the first week than other treatments (*P* < 0.01). As time increased, the growth rate of CK plant height slowed down, with the fastest growth rate being −0.49 MPa treatment in the second week, −1.02 MPa treatment in the third week, and −0.73 MPa treatment in the fourth week (Fig. 2C). For the total root length, the treatment with −0.04 MPa has the longest root length, which is 64.36, 11.7 cm longer than CK. With the decrease of environmental water potential, the total length of the root system showed a decreasing trend, with treatments of −0.29 MPa and below significantly lower than CK

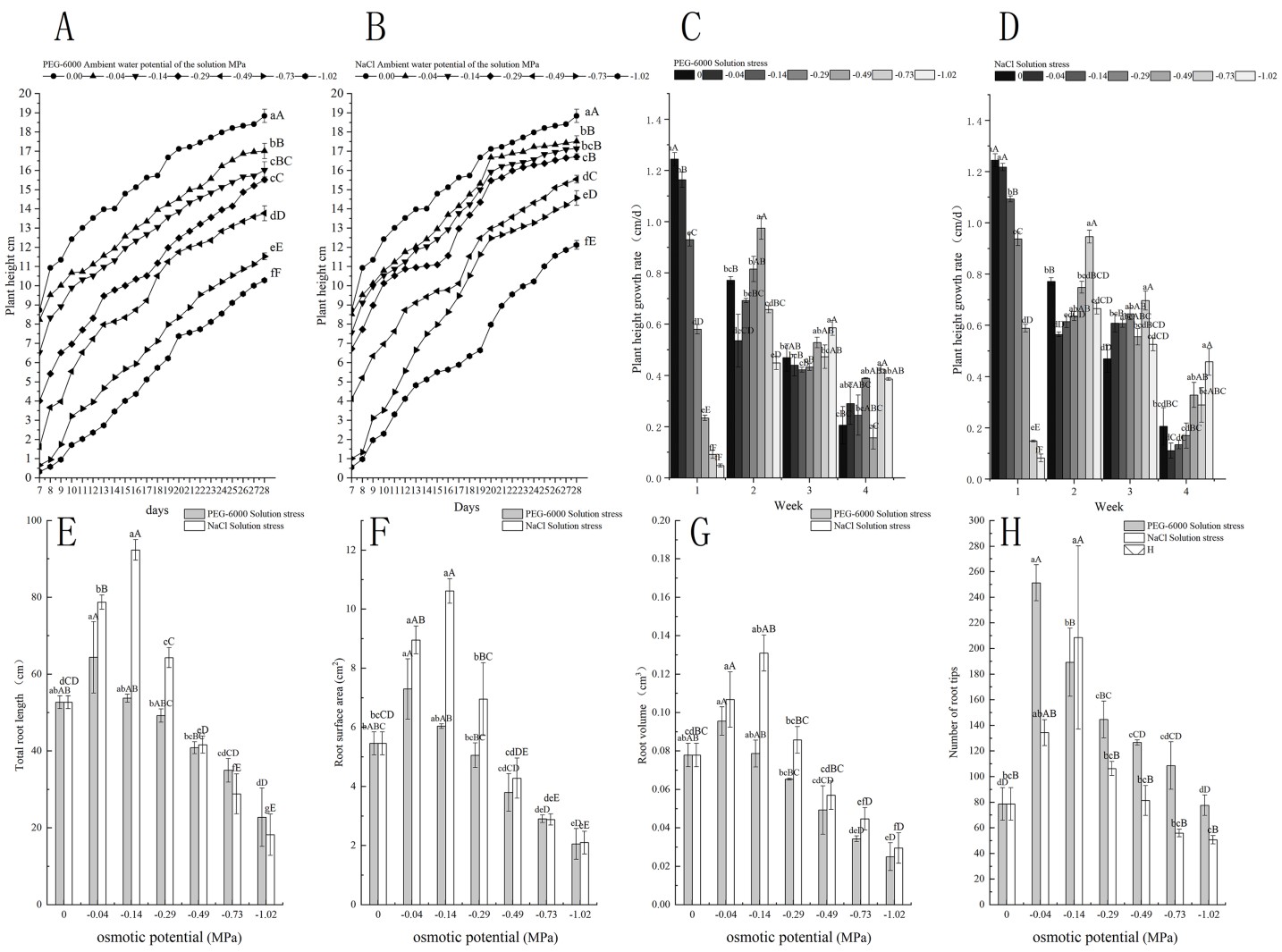

**Figure 2 Effects of processing on the growth of aboveground and underground parts.** Changes in plant height of *E. nutans* under PEG-6000 and NaCl solution stress (A, B), plant height trends (C, D), total root length (E), root surface area (F), root volume (G), and root tip number (H) on nutans. Different lowercase letters indicate significant differences in the same solution under different treatments ($P < 0.05$), different capital letters indicate that the differences in the same solution under different treatments reach a highly significant level ($P < 0.01$).

($P < 0.05$). When treated with −1.02 MPa, the total length of the root system was the lowest, at 22.78 cm (Fig. 2E). In terms of root surface area, the treatment with −0.04 and −0.14 MPa significantly increased the root surface area compared to CK ($P < 0.05$). As the environmental water potential decreased, the root surface area showed a decreasing trend, with the lowest being −1.02 MPa treatment, resulting in a root surface area of 2.05 cm² (Fig. 2F). In terms of root volume, the root volume of −0.04 and −0.14 MPa treatments was higher than that of CK. When treated with −0.29, −0.49, −0.73, −1.02 MP, the root volume showed a significant decrease trend compared to CK ($P < 0.05$). The minimum root volume is 0.02 cm³ after treatment with −1.02 MPa (Fig. 2G). In terms of the number of root tips, −0.04, −0.14, −0.29, −0, and 49 MPa treatments were significantly higher than CK ($P < 0.01$), with −0.04 treatment having the highest number of 251 (Fig. 2H).
Under simulated salinization stress with NaCl solution, the plant height of *E. nutans* showed a decreasing trend with the decrease of environmental water potential within 7–28 days (Fig. 2B). In terms of plant height growth rate, CK and −0.04 MPa were significantly higher in the first week than other treatments ($P < 0.01$). The plant height growth rate was the fastest in the −0.73 MPa treatment in the second and third weeks, and the growth rate was the fastest in the −1.02 MPa treatment in the fourth week (Fig. 2D). For the total length of the root system, the treatment with −0.14 MPa had the longest root length, which was 92.36 cm, showing a highly significant level of 39.7 cm longer than CK ($P < 0.01$); With the decrease of environmental water potential, the total root length showed a very significant decrease trend ($P < 0.01$), with the lowest being −1.02 MPa treatment, resulting in a total root length of 5.37 cm (Fig. 2E). In terms of root surface area, the root surface areas of −0.04, −0.14, and −0.29 MPa treatments were all greater than CK. The maximum root surface area of −0.14 MPa treatment was 10.61 cm$^2$, while the minimum surface area of −1.02 MPa treatment was 2.1 cm$^2$ (Fig. 2F). For the root volume, when treated with −0.04, −0.14, and −0.29 MPa, it was significantly higher than CK ($P < 0.05$), with a maximum root volume of 0.131 cm$^3$ under −0.14 MPa treatment. As the environmental water potential decreases, the root volume showed a downward trend, with a minimum of −1.02 MP treatment reaching 0.02 cm$^3$ (Fig. 2G). In terms of the number of root tips, −0.04, −0.14, and −0.29 MPa treatments were higher than CK, with −0.14 MPa treatment being significantly higher than CK ($P < 0.01$), with 134 root tips. The minimum processing time was −1.02 MP for 50 units (Fig. 2H).

## Effects of treatments on the content of C, N, and P in leaves and roots

Under simulated drought stress with PEG-6000 solution, the organic carbon content in leaves of CK treatment was lower than that of other treatments. Except for −0.04 MPa treatment, the organic carbon content was significantly higher than CK ($P < 0.05$), with the highest being −0.73 MPa treatment and the highest being 467.54 g·kg$^{-1}$. The −1.02 MPa treatment decreased by 4.64 g·kg$^{-1}$ compared to −0.73 MPa treatment (Fig. 3A). In terms of root organic carbon content, with the decrease of environmental water potential, the root organic carbon content showed a decreasing trend, and all treatments were significantly lower than CK ($P < 0.01$). The root organic carbon content of CK was 411.11 g·kg$^{-1}$. The minimum treatment pressure was −1.02 MPa, and the organic carbon content was only 286.59 g·kg$^{-1}$ (Fig. 3D). In terms of total nitrogen content in leaves, the treatments of −0.04, −0.14, and −0.29 MPa were significantly lower than CK ($P < 0.05$), while the treatments of −0.73 and −1.02 MPa were extremely significantly higher than CK ($P < 0.01$), with the lowest content of 18.58 g·kg$^{-1}$ in the −0.04 MPa treatment. The maximum total nitrogen content of −1.02 MPa treatment was 29.44 g·kg$^{-1}$ (Fig. 3B). In terms of total nitrogen content in the root system, the total nitrogen content in the root system decreases with the decrease of environmental water potential. The nitrogen content in the CK root system was 11.21 g·kg$^{-1}$ higher than that in the treatment group, and the lowest nitrogen content in the −1.02 MPa treatment was 7.07 g·kg$^{-1}$ (Fig. 3E). In the total phosphorus content of leaves, with the decrease of environmental water potential, the total phosphorus content of leaves showed a trend of first increasing and then decreasing.

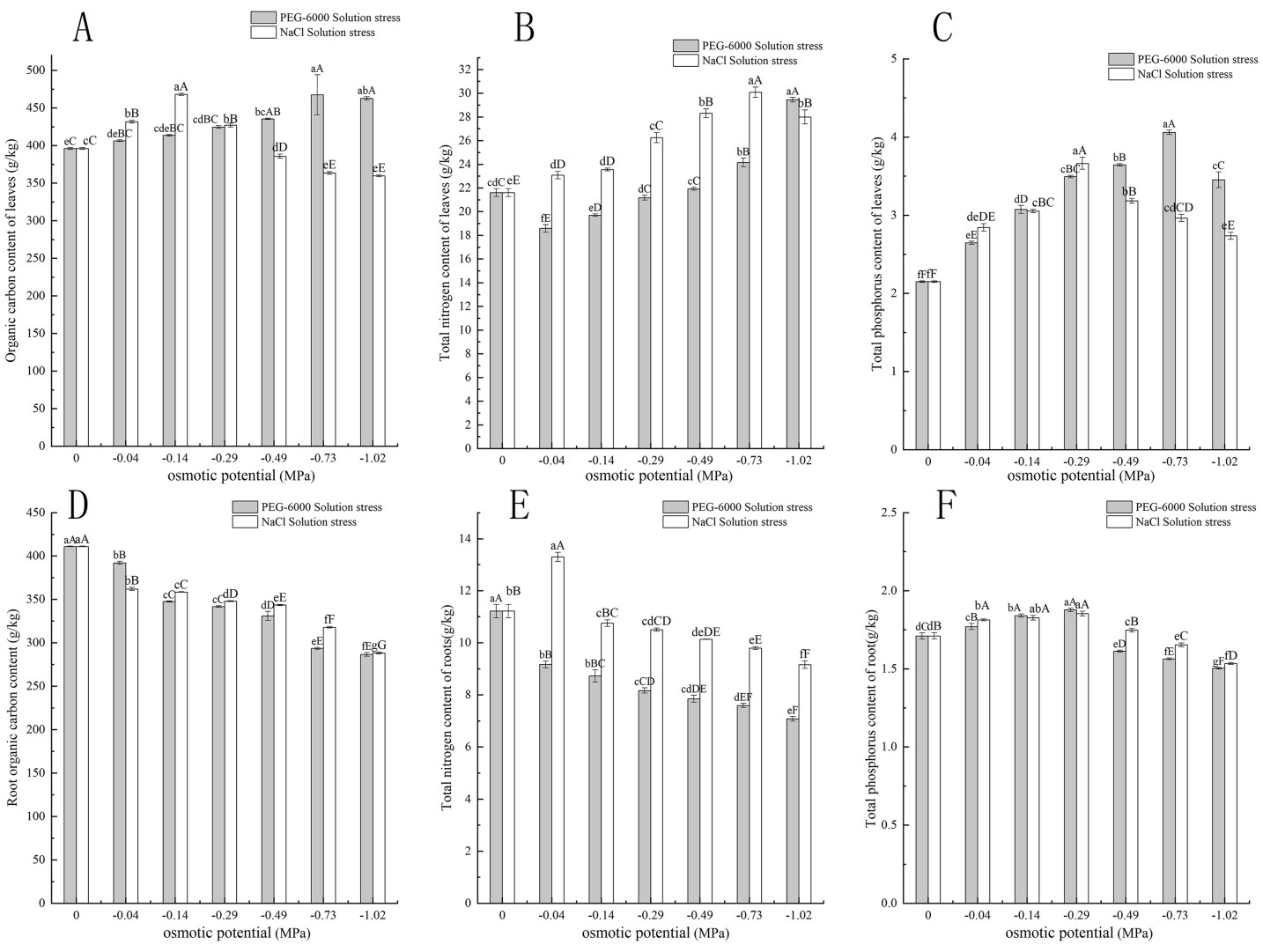

**Figure 3 Effect of processing on C, N, and P content in the aboveground and underground parts.** Two types of solution stress affect *E. nutans* leaf organic carbon (A), leaf total nitrogen (B), leaf total phosphorus (C), root organic carbon (D), root total nitrogen (E), and root total phosphorus (F) contents on nutans. Different lowercase letters indicate significant differences in the same solution under different treatments ($P < 0.05$), different capital letters indicate that the differences in the same solution under different treatments reach a highly significant level ($P < 0.01$).

The total phosphorus content of leaves in all treatments was significantly higher than CK ($P < 0.01$), with the highest content of 4.06 g·kg$^{-1}$ in the −0.73 MPa treatment (Fig. 3C). In terms of total phosphorus content in the root system, with the decrease of environmental water potential, the total phosphorus content in the root system showed a trend of first increasing and then decreasing. The highest content in the −0.29 MPa treatment was 1.87 g·kg$^{-1}$, and the −0.04, −0.14, and −0.29 MPa treatments were significantly higher than CK ($P < 0.01$). The −0.49, −0.73, and −1.02 MPa treatments were all significantly lower than CK ($P < 0.01$), and the lowest was 1.50 g·kg$^{-1}$ in the −1.02 MPa treatment (Fig. 3F).

Under simulated salinization stress with NaCl solution, the organic carbon content in leaves showed a trend of first increasing and then decreasing. Among them, the organic carbon content in leaves under −0.04, −0.14, and −0.29 MPa treatments was significantly higher than CK ($P < 0.01$), but with the decrease of environmental water potential, the organic carbon content in leaves under −0.49, −0.73, and −1.02 MPa treatments was significantly lower than CK ($P < 0.01$) (Fig. 3A). In terms of root organic carbon, with the decrease of environmental water potential, all treatments were significantly lower than CK ($P < 0.01$) and showed a very significant downward trend ($P < 0.01$), with the lowest being −1.02 MPa treatment, and the organic carbon content was only 288.29 g·kg$^{-1}$ (Fig. 3D). In terms of total nitrogen content in leaves, with the decrease of environmental water potential, the total nitrogen content in leaves of all treatments was significantly higher than CK ($P < 0.01$). Among them, the −0.73 MPa treatment had the highest total nitrogen content of 30.09 g·kg$^{-1}$, and the −1.02 MPa treatment decreased by 2.01 g·kg$^{-1}$ compared to the −0.73 MPa treatment (Fig. 3B). In terms of root total nitrogen, the −0.04 MPa treatment had a root total nitrogen content of 13.29 g·kg$^{-1}$, which was significantly higher than the other treatments of C ($P < 0.01$). The root total nitrogen content of −0.14, −0.29 MPa, −0.49, −0.73, and −1.02 MPa treatments was significantly lower than that of CK ($P < 0.05$), with the lowest content of 9.15 g·kg$^{-1}$ in the −1.02 MPa treatment (Fig. 3E). In the total phosphorus content of leaves, with the decrease of environmental water potential, the total phosphorus content of leaves showed a trend of first increasing and then decreasing. The total phosphorus content of leaves in all treatments was significantly higher than CK ($P < 0.01$), with the highest content of 3.66 g·kg$^{-1}$ in the −0.29 MPa treatment (Fig. 3C). In the total phosphorus content of the root system, with the decrease of environmental water potential, the total phosphorus content of the root system first increases and then decreases. The highest content of 0.29 MPa treatment was 1.87 g·kg$^{-1}$, and the treatments of −0.04, −0.14, and −0.29 MPa were all significantly higher than CK ($P < 0.01$). The treatments of −0.49, −0.73, and −1.02 MPa were all significantly lower than CK ($P < 0.01$), and the lowest was 1.53 g·kg$^{-1}$ under −1.02 MPa treatment (Fig. 3F).

**Effects of treatments on C:N, N:P of seedlings**

Under simulated drought stress with PEG-6000 solution, in terms of leaf C:N, with the decrease of environmental water potential, leaf C:N showed a trend of first increasing and then decreasing. Among them, the maximum C:N value under −0.04 MPa treatment was 21.87, and the minimum value under −1.02 MPa treatment was 15.72. Moreover, under −1.02 MPa treatment, it was significantly lower than CK ($P < 0.01$) (Fig. 4A). In terms of root C:N, the root C:N of each treatment was significantly higher than that of CK ($P < 0.05$), with the highest being 42.76 in the −0.04 MPa treatment (Fig. 4C). In terms of leaf N:P, the N:P values of all treatments were significantly lower than those of CK ($P < 0.01$), with the lowest N:P value of 5.94 in the −0.73 MPa treatment (Fig. 4B). In terms of root N:P, the N:P values of each treatment were significantly lower than those of CK ($P < 0.01$), with the lowest being −0.29 MPa treatment, 4.34 (Fig. 4D).

Under simulated salinization stress with NaCl solution, with the decrease of environmental water potential, the C:N value of leaves showed a trend of first increasing

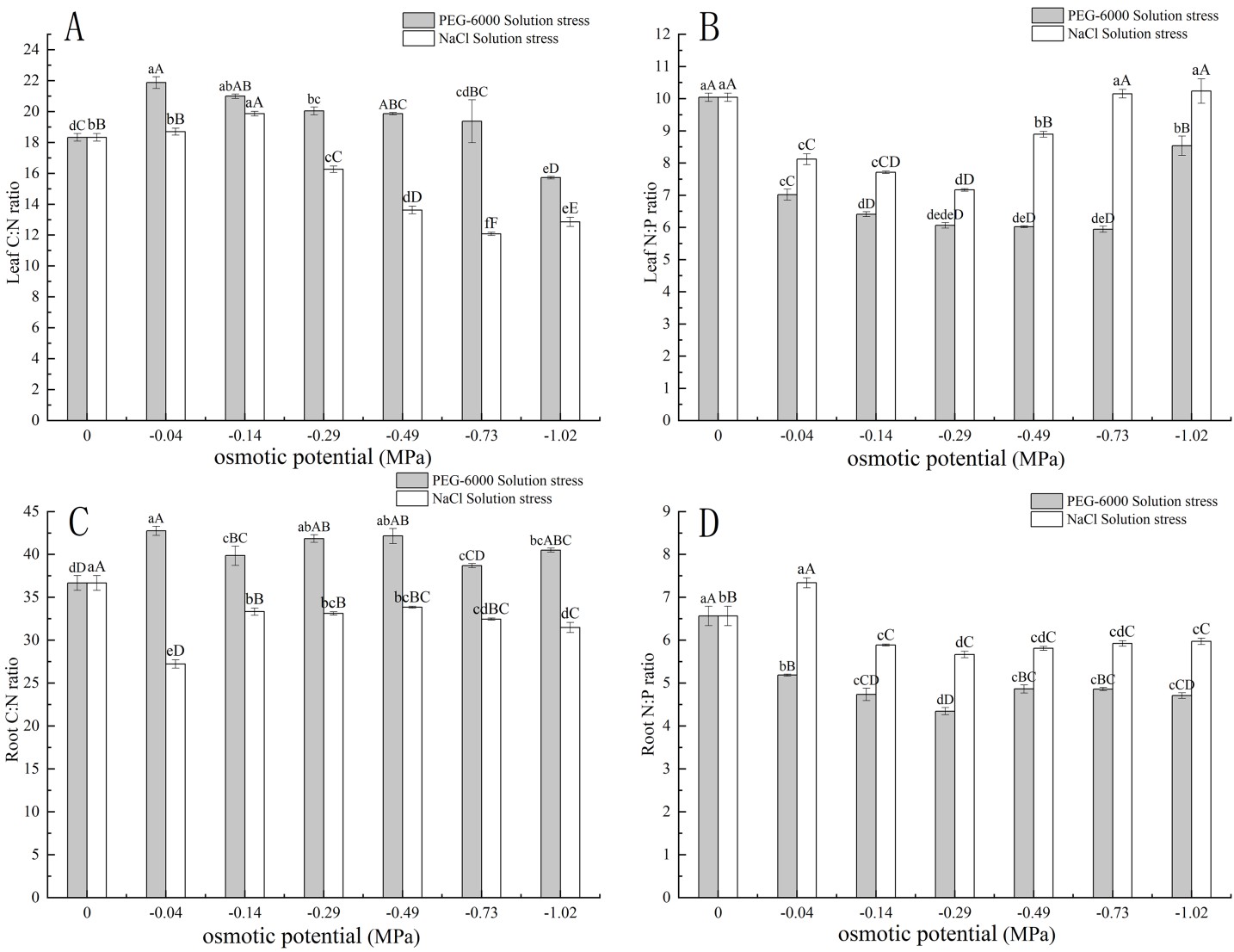

**Figure 4 Effects of two solutions on C:N and N:P in leaves and roots.** Two types of solution stress on *E. nutans* leaves C:N (A), leaf N:P (B), root C:N (C), and root N:P (D). Different lowercase letters indicate significant differences in the same solution under different treatments ($P < 0.05$), different capital letters indicate that the differences in the same solution under different treatments reach a highly significant level ($P < 0.01$).

and then decreasing. Among them, the maximum C:N value under −0.14 MPa treatment was 19.86, and the minimum value under −1.02 MPa treatment was 12.72. Among them, the values under −0.29, −0.49, −0.73, and −1.02 MPa treatment were significantly lower than CK ($P < 0.01$) (Fig. 4A). In terms of root C:N, the root C:N of each treatment was significantly lower than that of CK ($P < 0.01$), with the lowest being 27.23 in the −0.04 MPa treatment (Fig. 4C). In terms of leaf N:P, the N:P values of −0.04, −0.14, −0.29, and −0.49 MPa treatments were significantly lower than CK ($P < 0.01$), while the N:P values of −0.73 and −1.02 MPa treatments were slightly higher than CK, with the highest being 10.23 under −1.02 MPa treatment (Fig. 4B). In terms of root N:P, the N:P value of −0.04 MPa treatment was 7.33, which was significantly higher than CK ($P < 0.01$), while the N:P values

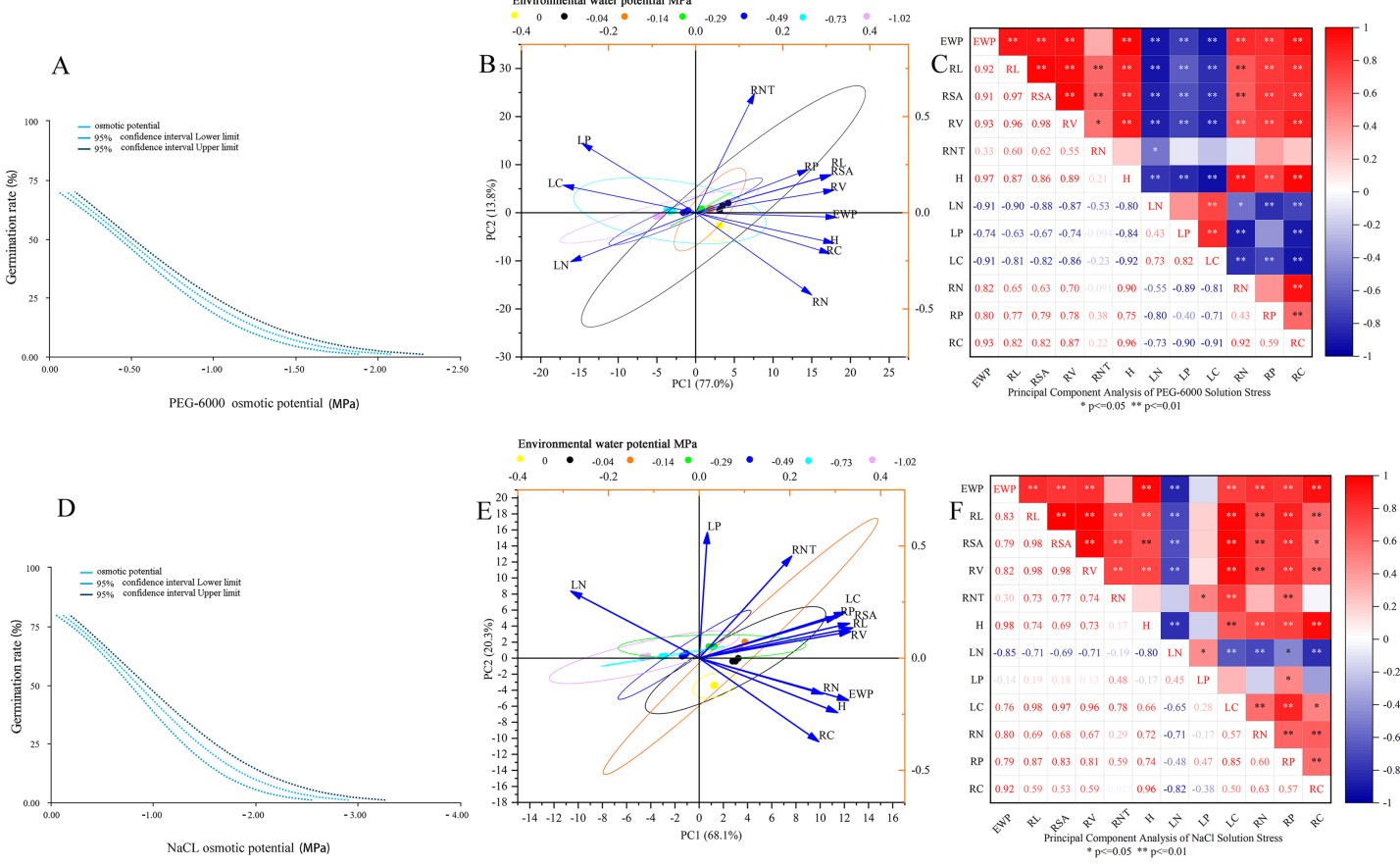

**Figure 5 Comprehensive analysis.** Dose response analysis of PEG-6000 stress on seed germination (A); principal component analysis of PEG-6000 on seedling growth (B); factor analysis of PEG-6000 on seedling growth (C). Dose response analysis of NaCl stress on seed germination (D); principal component analysis of NaCl on seedling growth (E); factor analysis of NaCl on seedling growth (F). (EWP, environmental water potential; RL, total root length; RSA, root surface area; RV, root volume; RNT, number of root tips; H, plant height; LN, total nitrogen content in leaves; LP, total phosphorus content in leaves; LC, organic carbon content in leaves; RN, total nitrogen content in roots; RP, total phosphorus content in roots; RC, organic carbon content in roots). **$P < 0.01$, *$P < 0.05$.

of other treatments were significantly lower than CK ($P < 0.01$). Among them, the lowest N:P value of −0.29 MPa treatment was 5.66 (Fig. 4D).

## Comprehensive evaluation

As shown in Fig. 5A, the environmental water potential of PEG-6000 solution showed a linear relationship with germination rate ($p < 0.001$). The model formula was Probit ($p$) = 0.670 + 1.442 * environmental water potential (PEG-6000), and the LD$_{50}$ value was −0.465 (95% CI [0.425–0.505]). According to Figs. 5B and 5C, it can be seen that under PEG-6000 stress, environmental water potential showed a highly significant negative correlation with leaf organic carbon content, leaf total nitrogen content, and leaf total phosphorus content ($P < 0.01$). The environmental water potential showed a highly significant positive correlation with plant height, total root length, root surface area, root volume, root organic carbon, root total nitrogen content, and root total phosphorus content ($P < 0.01$). There was a highly significant negative correlation between plant height

and leaf organic carbon content, leaf total nitrogen content, and leaf total phosphorus content ($P < 0.01$). There was a highly significant positive correlation between plant height and total root length, root surface area, root volume, root organic carbon, root total nitrogen content, and root total phosphorus content ($P < 0.01$). The total root length, root surface area, and root volume showed a highly significant negative correlation with leaf organic carbon content, leaf total nitrogen content, and leaf total phosphorus content ($P < 0.01$). The total root length, root surface area, root volume showed a highly significant positive correlation with plant height, root organic carbon, root total nitrogen content, and root total phosphorus content ($P < 0.01$).

As shown in Fig. 5D, there was a linear relationship between the environmental water potential of NaCl solution and germination rate ($p < 0.001$), with Probit ($p$) = 0.961 + 1.131 * environmental water potential (NaCl), corresponding to an $LD_{50}$ value of −0.850 MPa (95% CI [0.770–0.929]). Under the simulated salinization stress of NaCl solution (Figs. 5E and 5F), the environmental water potential showed a negative correlation with the total nitrogen content and total phosphorus content of leaves, among which it reached a very significant level with the total nitrogen content of leaves ($P < 0.01$); The environmental water potential showed a highly significant positive correlation with leaf organic carbon content, total root length, root surface area, root volume, plant height, root organic carbon content, root total nitrogen content, and root total phosphorus content ($P < 0.01$). There was a highly significant negative correlation between plant height and total nitrogen content in leaves ($P < 0.01$). There was a highly significant positive correlation between plant height and total root length, root surface area, root volume, leaf organic carbon, root organic carbon, root total nitrogen content, and root total phosphorus content ($P < 0.01$). The total root length, root surface area, root volume, and total nitrogen content in leaves showed a highly significant negative correlation ($P < 0.01$). The total root length, root surface area, root volume showed a significant positive correlation with plant height, leaf organic carbon content, root organic carbon, root total nitrogen content, and root total phosphorus content ($P < 0.05$).

# DISCUSSION

## Effect of drought stress and salinity stress on seed germination of *E. nutans*

Seed germination is the starting point of spermophyte life history, and water is one of the necessary conditions for seed germination. Water deficiency will affect the activity of internal enzymes, cell division, and other physiological metabolic processes in seeds. At the same time, water deficiency may cause seeds to lose vitality due to not being able to absorb enough water at once, thereby affecting seed germination (*Costa et al., 2021*; *Atia et al., 2011*; *Ahmad & Li, 2021*). This experiment showed that water deficit caused by PEG-6000 simulated drought stress and NaCl simulated salinization stress can affect the germination of *E. nutans* seeds (Fig. 1A). Under −0.04 MPa treatment, NaCl solution stress promoted *E. nutans* seed germination, which was 2.77% higher than CK. PEG-6000 solution stress inhibited *E. nutans* seed germination; Moreover, under the lowest environmental water

**Table 1 Affiliation function values and ranking of seed germination data.**

| Solution | Membership function value | | | | | | Average value | Sort |
|---|---|---|---|---|---|---|---|---|
| | G | GE | Bud length | Root length | Germination index | Vigor index | | |
| PEG-6000 | 0.50 | 0.52 | 0.52 | 0.44 | 0.63 | 0.59 | 0.52 | 2 |
| NaCl | 0.64 | 0.57 | 0.62 | 0.46 | 0.62 | 0.46 | 0.56 | 1 |

potential of −1.02 MPa in this experiment, the germination rate of NaCl solution treatment still exceeded 40%, while the germination rate of PEG-6000 solution treatment was only 26%. According to the dose-response analysis results, the $LD_{50}$ values under PEG-6000 and NaCl stress were −0.465 and −0.850 MPa, respectively. This may be due to the different expression of ABA signals in drought and salt stress during the germination process of *E. nutans* seeds, or because *E. nutans* has relatively high salt tolerance, and the PEG-6000 solution stress treatment has a more direct impact on the seed germination process (*He et al., 2022*; *Li et al., 2020*; *Nakashima & Yamaguchi-Shinozaki, 2013*). Under all treatments of the two solutions, GE, GI, and VI were lower than CK (Figs. 1B, 1E and 1F), which may be related to the response of seed enzyme activity to water deficiency. Although both solutions reduced GE, GI, and VI under stress, under the same environmental water potential treatment, the GE, GI, and VI of NaCl solution were higher than those of PEG-6000 solution treatment. This may be due to the PEG-6000 solution treatment affecting the process of seed germination, causing the seeds to not fully absorb water and resulting in slow germination speed (*Dietz, Zörb & Geilfus, 2021*; *Pan et al., 2021*). The study by *Wang, Wu & Yu (2017)* showed that with the increase of PEG-6000 and NaCl solution concentration, the germination index of *E. nutans* seeds showed a downward trend, which is basically consistent with this study.

Under the same environmental water stress of the two solutions in this study (Figs. 1C and 1D), *E. nutans* bud length showed a significant decrease, and each treatment was highly significant lower than the CK treatment ($P < 0.01$), but the decrease in bud length was lower under the NaCl solution treatment than the PEG-6000 solution treatment; The bud length of PEG-6000 treatment was higher than that of NaCl treatment when the ambient water potential was −0.04, −0.14 and −0.29 MPa treatments; the bud length after NaCl treatment exceeded that of PEG-6000 when the ambient water potential was −0.490, 0.73, and −1.02 MPa treatments. PEG-6000 and NaCl solution treatments also reduced root length, and root length was greater under all PEG-6000 treatments than NaCl treatments. This is due to the fact that under PEG-6000 solution stress *E. nutans* safeguards root water uptake by increasing root biomass, whereas NaCl solution causes direct damage to plant root cells inhibiting plant root growth (*Dietz, Zörb & Geilfus, 2021*; *Pan et al., 2021*). It has been shown (*Song, Yang & Jing, 2022*; *Wang, Liu & Zhang, 2021*) that the root length and bud length of *E. nutans* seeds show a decreasing trend when stressed by PEG-6000 and NaCl solutions during germination, which is basically consistent with the present study. Moreover, the analysis of seed germination indexes under PEG-6000 and NaCl stresses by the affiliation function method revealed that

(Table 1), the inhibition of *E. nutans* seed germination by PEG-6000 solution stress under the same environmental water potential was stronger than that by NaCl solution stress.

## Effects of drought stress and salinity stress on *E. nutans* plant height, total root length, root surface area, and root volume

The most intuitive response of plants to water deficiency is the morphological indicators of the aboveground and underground parts. Water deficiency can lead to shorter plants, thinner stems and leaves, and also affect the total length, surface area, volume, and other aspects of the root system (*Ulrich et al., 2022*; *Dell'Aversana et al., 2021*; *Duan, Sebastian & Dinneny, 2015*). This experimental study showed that with the decrease of environmental water potential, the stress of two different solutions would reduce the plant height of *E. nutans*. As the environmental water potential decreased, the stress of two different solutions reduced the plant height of *E. nutans*. However, the treatment with PEG-6000 solution at −0.04, −0.14, and −0.29 MPa showed a significant decrease trend, while the treatment with NaCl solution did not show this situation. The treatment effects of two solutions showed the same decrease trend at −0.49 MPa. However, with the decrease of environmental water potential, the growth rate of plant height under both solution stresses showed an increase in low environmental water potential over time, while a decrease in high environmental water potential over time. The reason may be that during the first two weeks after seed germination under high environmental water potential, the growth rate of plant height is faster than under low concentration environmental water potential stress. As time goes on, the growth rate of plant height under low concentration environmental water potential gradually increases, which is due to the slowing down of plant height growth under high concentration environmental water potential. This is basically consistent with *Ling et al. (2020)*'s research on rice seedlings under salt stress.

When lacking water, plants compensate for the decrease in root absorption area by enhancing root vitality and maintaining a higher root biomass, thereby maintaining a higher root water absorption capacity. The total length, surface area, volume, and number of root tips of the root system have a significant impact on the water absorption capacity of the plant root system (*Bloom, 2015*; *Fowdar et al., 2022*; *Lambers, Martinoia & Renton, 2015*). This experimental study showed that with the decrease of environmental water potential, the total root length of *E. nutans* showed a trend of first increasing and then decreasing. Among them, under the treatment of PEG-6000 solution, the total root length of −0.04 Mpa treatment was the longest, while under the treatment of NaCl solution, the total root length of −0.14 MPa treatment was the longest. Moreover, under the treatment of −0.04, −0.14, −0.29, and −0.49, the total root length of NaCl solution treatment was greater than that of PEG-6000 solution treatment, but under the treatment of −0.73 and −1.02 MPa, the total root length of NaCl was significantly reduced, treatment below PEG-6000. The surface area, volume, and total length of the root system showed identical trends, and the maximum area and volume of PEG-6000 and NaCl solutions were also −0.04 and −0.14 MPa treatments, respectively. However, in terms of surface area and volume, NaCl treatments were higher than PEG-6000. The number of root tips also showed a trend of first increasing and then decreasing. The PEG-6000 solution with −0.04 MPa treatment

had significantly higher root tip numbers than other treatments ($P < 0.01$), and the NaCl solution with −0.14 MPa treatment was also significantly higher than other treatments ($P < 0.01$). However, the PEG-6000 solution had a relatively gentle decrease in root tip numbers, while the NaCl solution had a significant decrease in root tip numbers, which was lower than the CK treatment at −0.73 and −1.02 MPa treatments. Related studies have shown (*Hazman & Brown, 2018*; *Wang et al., 2018*; *Chaves et al., 2002*) that a small amount of water deficiency promotes plant root growth, resulting in an increase in the total length, surface area, volume, and number of root tips of plant roots. However, as the environmental water potential decreases, the degree of drought and salt damage to the roots increases, leading to root growth inhibition, which is basically consistent with the results of this study.

## Effects of drought stress and salinity stress on the organic carbon, total nitrogen and total phosphorus contents of *E. nutans* leaves and roots

When facing drought and salt stress, the organic carbon content of plant leaves and roots is affected to adapt to the impact of environmental water potential changes on plant growth (*Ulrich et al., 2022*; *Dell'Aversana et al., 2021*; *Priya et al., 2021*). This study indicated that the organic carbon, total nitrogen, and total phosphorus contents of *E. nutans* leaves were all affected by the decrease in environmental water potential. In terms of organic carbon content, PEG-6000 solution showed a gradual upward trend under stress, and the highest content was observed when treated with −0.73 MPa. Under NaCl solution stress, the content showed a trend of first increasing and then decreasing, with the highest content observed under −0.14 MPa treatment. The organic carbon content in the root system of *E. nutans* was directly affected by environmental water potential. With the decrease of environmental water potential, both PEG-6000 solution and NaCl solution showed a significant decrease trend ($P < 0.01$), and except for the −0.04 MPa treatment, the organic carbon content in the root system under NaCl solution stress was higher than that under PEG-6000 solution stress. As the degree of water deficiency increases, the organic carbon content in the roots of *E. nutans* showed a decreasing trend, which is different from the results of *Dell'Aversana et al. (2021)* under salt stress in barley and may be due to different research subjects.

Nitrogen (N) and phosphorus (P) are both basic macronutrients that limit plant growth and primary productivity in different terrestrial ecosystems (*Bloom, 2015*; *Fowdar et al., 2022*; *Lambers, Martinoia & Renton, 2015*). In terms of leaves, the effects of PEG-6000 and NaCl stress on the total nitrogen content of *E. nutans* showed completely different trends. Under salt stress, the total nitrogen content of leaves was higher than CK, but under drought stress, it showed a trend of first decreasing and then increasing. Both solutions of stress increased the total phosphorus content in *E. nutans* leaves, and both showed a trend of first increasing and then decreasing. In terms of roots, under PEG-6000 stress, the total nitrogen content of *E. nutans*' roots was lower than that of CK, while under the treatment of −0.04 MPa in NaCl solution, the total nitrogen content of *E. nutans*' roots was higher than that of CK, and under the same environmental water potential, the total nitrogen

content of *E. nutans*' roots was higher than that under PEG-6000 solution stress. The decrease in environmental water potential had little impact on the total phosphorus content in roots, and both solutions showed a trend of first increasing and then decreasing under stress. The trend of nitrogen content changes in plants in this experiment is basically consistent with the study of *Zhao et al. (2020)* on *E. nutans* under drought stress. The changes in phosphorus content are consistent with the research results of *Liang et al. (2022)*, which may be due to different experimental environments and methods.

## Effects of drought stress and salinity stress on C:N and N:P in leaves and roots of *E. nutans*

C and N are the two most fundamental elements for plant growth and development, and their mutual coupling makes C:N an important indicator for exploring plant element allocation and adaptation strategies (*He et al., 2019*; *Wang et al., 2015*). For example, high C:N plants have high nitrogen use efficiency; Low C:N litter has the characteristic of fast decomposition. This experiment showed that water deficit affected C:N of *E. nutans* leaves and roots, basically showing a rise followed by a fall, and that C:N was higher for PEG-6000 simulated drought stress than for NaCl simulated salinity stress under the same environmental water potential treatment. It may indicate that water potential stress had an effect on the nitrogen utilization of *E. nutans* and showed a trend of first had and then decreasing, and NaCl solution stress at the same ambient water potential had a greater effect on nitrogen utilization, which is consistent with the study of *Wan et al. (2022)*.

The balance of the coupling of N and P has a great impact on the structure and function of plants from the molecular level to the biome level or different biological tissues (*Niu et al., 2019*; *Liang et al., 2022*). This experiment showed that water deficit affected the nitrogen and phosphorus balance of *E. nutans* leaves and roots, and showed a trend of first will then increase in the leaves nitrogen and phosphorus ratio. Both leaves and roots N:P were higher in NaCl solution treatment than in PEG-6000 solution treatment under the same ambient water potential treatment. It may be suggested by the N:P aspect that NaCl solution stress has a greater effect on the biological organization and function of *E. nutans*.

## Effects of C, N and P contents of above and below ground parts on the growth of *E. nutans*

Drought stress and salt stress alter the C, N, and P contents of plant leaves, and the changes in plant leaf and root morphology are particularly pronounced (*Zhou et al., 2022*; *Dibar et al., 2020*).

The results of the correlation analysis showed (Figs. 5C and 5F) that the growth of plant leaves and roots under drought stress would be hindered by the reduction of leaf organic carbon, total nitrogen and total phosphorus contents. The main factor leading to the stunted growth of plant leaves and roots under salinity stress is the total nitrogen content of the leaves. The main factor affecting plant growth under the two solution stresses is the change of elements in the leaves, so it may be possible to improve the resistance of forage seedlings to drought stress and salt stress by foliar spraying of fertilizers (*Gao et al., 2022*; *Shabbir et al., 2016*). The results of the analysis of the growth indicators of *E. nutans*

**Table 2 Seedling growth data affiliation function values and ranking.**

| Solution | Membership function value | | | | | | | | | | | Average value | Sort |
|---|---|---|---|---|---|---|---|---|---|---|---|---|---|
| | RL | RSA | RV | RNT | H | LN | LP | LC | RN | RP | RC | | |
| PEG-600 | 0.51 | 0.45 | 0.49 | 0.32 | 0.49 | 0.35 | 0.55 | 0.33 | 0.34 | 0.50 | 0.46 | 0.43 | 2 |
| NaCl | 0.50 | 0.45 | 0.46 | 0.31 | 0.56 | 0.48 | 0.50 | 0.45 | 0.36 | 0.59 | 0.48 | 0.46 | 1 |

seedlings using the affiliation function method showed (Table 2) that under the same environmental water stress, *E. nutans* seedlings were subjected to salinity stress to a lesser extent than drought stress and showed some salt tolerance.

# CONCLUSIONS

This experiment unified the effects of PEG-6000 and NaCl solutions on water deficit in forage by adjusting the osmotic potential of the solution, and explored the effects of drought stress and salinization stress on the growth of *E. nutans* seedlings under the same environmental water potential stress. The results indicated that although even under the same environmental water potential, there was a significant difference in the effects of drought stress and salinization stress on the germination of *E. nutans* seeds, and the inhibition of drought stress on seed germination was stronger than that of salt stress. This difference was further reflected in the normal growth of the aboveground and underground parts of the seedlings, as well as changed in the content of organic carbon, total nitrogen, and total phosphorus. A comprehensive assessment of *E. nutans* seed germination and seedling indicators using the affiliation function method revealed that *E. nutans* was more severely affected by drought stress at both the seed germination and seedling growth stages, showing results of better salt tolerance than drought tolerance.

## Funding

This work was funded by the Central Guiding Local Science and Technology Development Fund Project (ZYYD2023000147), the National Natural Science Foundation projects (U21A20240; U20A2050), and the Construction of the "New Agricultural Science" Plateau Plant Production Professional Improvement Practice and Innovation Ability Platform (Zangcai Yuzhi 2023-1). The funders had no role in study design, data collection and analysis, decision to publish, or preparation of the manuscript.

## Grant Disclosures

The following grant information was disclosed by the authors:
Central Guiding Local Science and Technology Development Fund Project: ZYYD2023000147.
National Natural Science Foundation Projects: U21A20240, U20A2050.

Construction of the "New Agricultural Science" Plateau Plant Production Professional Improvement Practice and Innovation Ability Platform: 2023-1.

## Competing Interests

The authors declare that they have no competing interests.

## Author Contributions

- Jianting Long conceived and designed the experiments, performed the experiments, analyzed the data, prepared figures and/or tables, complete thesis revision and writing, and research for thesis details, and approved the final draft.
- Mengjie Dong performed the experiments, authored or reviewed drafts of the article, and approved the final draft.
- Chuanqi Wang analyzed the data, authored or reviewed drafts of the article, and approved the final draft.
- Yanjun Miao performed the experiments, authored or reviewed drafts of the article, and approved the final draft.

## Data Availability

The raw data is available in the Supplemental Files.

## Supplemental Information

Supplemental information for this article can be found online at http://dx.doi.org/10.7717/peerj.15968#supplemental-information.

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
