# Peer review of "Effects of drought and salt stress on seed germination and seedling growth of Elymus nutans"

_PeerJ, doi:10.7717/peerj.15968_

## Round 0.1 · original submission · Major Revisions

The English language and format of the writing need to be significantly improved and explained all the comments raised by all the reviewers.

Reviewer 1 ·

Basic reporting

Due to author's English proficiency, including problems in grammar, wording, and sentences, the manuscript was not well written.
The subtitles of the manuscript are not concise and descriptions of the results are wordy.
Figures are basically relevant to the content of the article, but the resolution is low. The captions of Figure 1 and 2 are not appropriately described.

Experimental design

The research question and the knowledge gap were not clearly defined and identified. The goals of the study wre ambiguous.
Both PEG6000 and NaCI stresses with different osmotic potentials are dose-response tests. One way ANOVA is not as good as dose-response analysis.
Factor analysis is not a proper method for this research, and the principal component analysis (upper part of Figure 5) does not contribute to the interpretation of the results.

Validity of the findings

Conclusions are needed to refine to conform with research question and goals.

Additional comments

Detailed description are needed for Materials and Methods
Why two separated germination tests were conducted (Petri dish with filtered paper and germination box with sand and Hoagland)?. Which data set was used to draw Figure 1 and results description?
Discussion section should not be the re-description of result parts. More attention needs to paid to interpretation of the results and comparison with literature.
For more specific errors, questions and suggestions, see annotation in the PDF manuscript.

Annotated reviews are not available for download in order to protect the identity of reviewers who chose to remain anonymous.

·

Basic reporting

At first, I would to mention the actual topic of the study – to evaluate to impact of different abiotic stress on first development stages of Elymus nutans – seed germination seedling.
Introduction section is very brief and doesn’t cover some previous studies on the topic (E. nutans salt tolerance):
• Growth of Elymus nutans in saline saline-alkali soil amended with calcium silicate slag: Performance and mechanism https://doi.org/10.11686/cyxb2020319
• Effects of Exogenous GABA on Seed Germination and Physiological Characteristics of Elymus nutans under NaCl Stress https://doi.org/10.11733/j.issn.1007-0435.2022.02.018
• Effects of salt, temperature and their interaction on seed germination and seedling growth of Elymus nutans https://www.cabdirect.org/cabdirect/abstract/20143223856
• Drought tolerance of wild Elymus nutans during germination and seedling establishment. https://doi.org/10.11686/cyxb2020396
• Comparative study on drought resistance of six native Elymus L. species seedlings. https://www.cabdirect.org/cabdirect/abstract/20173057895
The formatting of manuscript requires multiple improvements – there are missing spaces, commas, different font size, missing italic etc.
L51, L54: repeating, please rephrase
L238. L239: repeating, please rephrase

Experimental design

Some details of the study need to be clarified:
• L80: what “2.2.1.1000 g” means?
• watering (irrigation) regime is unclear
• does any fertilizers or nutrient sources were used?
Authors didn’t report the scale (amount) of the effect: L121 “significantly lower”, L124, L126 “decreased significantly”, L161 “significantly higher” and so on.
L112: details of the data analysis required

Validity of the findings

The Conclusion section is written vaguely. It’s interesting to compare the different methods of salt stress, but interpretation is strongly required. Some methods might be obtained from the study https://doi.org/10.1093/jpe/rtaa070
My notes about the main text:
• abbreviation “CR” is misleading
• References list has missing idents

Reviewer 3 ·

Basic reporting

This study investigated the effects of drought and salinity stress on seed germination and seedling growth of Elymus nutans. The experiment was well-designed and a large amount of data was collected. However, the English language and format of the writing need to be significantly improved. The statistical analyses could also be improved, particularly the mean separation method. Finally, the interpretation of Figure 5 needs to be supplemented.

Experimental design

1. Lines 20: English language needs to be improved. ‘The reduction of only leaf total nitrogen content’ should be a result of this study, how could one result affect other results? Besides, what do authors mean by ‘only leaf total nitrogen content’, do you mean total nitrogen in a single leaf?
2. Line 32: are the main reasons
3. Line 33: drought, water shortage, and soil salinization limit the sustainable development
4. E. nutans, the scientific name needs to be italic, please correct the whole manuscript.
5. Lines 44-45: English language needs to be improved.
6. Line 64: What do authors mean by ‘the test was sterilized and set aside’?
7. Please add space after each dot.
8. Line 93: Do authors mean the number of seeds germinated within 1 week?
9. Lines 100-110: Please check the font of this paragraph.
10. Line 108: Please cite the reference for the Kjeldahl method, molybdenum blue ascorbic acid method, and Elementar vario TOC elemental analysis.
11. In figure 1, it is not necessary to present mean separation by different alpha values. Instead, authors can compare drought and salinity stress under the same osmotic potential. Similar problems in figures 2-4.
12. In materials and methods, authors didn’t mention the measurement method for germination potential, bud length, and vitality index (Figure 1). Similarly, please be consistent on expression of the parameters presented in the figures, materials and methods and results.

Validity of the findings

13. Please add a dot when a sentence is finished.
14. Line 255: Since authors did comparation between PEG-6000 and NaCl treatments, mean separation between drought and salinity is necessary as I advised in comment #11.
15. Please consistently use either figure or fig. in results and discussion.
16. I am surprised the PCA and correlation results presented in figure 5 was not or only partially mentioned in materials and methods, not mentioned in results. However, it appears in the abstract and discussion. Please explained the methods used for the PCA and correlation analysis in more detail and interpret figure 5 in results.

---

## Round 0.2 · Minor Revisions

Thank you for the revised manuscript. Still, we have a few minor comments to improve the manuscript.

Reviewer 1 ·

Basic reporting

The authors have made a significant modification to the manuscript, but some language problems are still in it.
The way of some literature citation in the discussion section does not conform to academic norms.
In references section, names of some journals were not correctly written, and format of some journal name are inconsistent with others.
For specific suggestions, please see annotations in PDF manuscript.

Experimental design

Original primary research is within Aims and Scope of the journal.
The research question is clearly defined. The knowledge gap being investigated is identified. Methods are described with sufficient information to be reproducible by another investigator.

Validity of the findings

All underlying data have been provided; they are robust, statistically sound, & controlled.
Conclusions are well stated, linked to original research question & limited to supporting results.

Annotated reviews are not available for download in order to protect the identity of reviewers who chose to remain anonymous.

·

Basic reporting

I would to thank authors for the efforts to improve the manuscript, but it still requires multiple corrections. The formatting of manuscript has flaws mentioned previously (missing spaces, different fonts, etc.); please take time to correct all the issues. Some cases to mention:
• L413, L430: parts 4.4 and 4.5 have the same title?
• Missing spaces at L152, L168, L193, L344 and so on.

Experimental design

Some details of the study need to be clarified:
• L272: p=0 ?
• L126: please mention the settings of Origin Pro were used and refer to the raw data in Supplementary (spreadsheets files).
• L130: the details about dose-response meta-analysis are required (software, settings, raw data)
• Dots’ colors at PCA diagram (Figure 5) are similar and hard to interpret while they are located very close (yellow for 0; light green for -0.29; blue for -0.49 and purple for -1.02).

Validity of the findings

The Conclusion section is still needs improvement. Please summarize the Discussion section to highlight main point of the study.
L303-L306: better fits to the Intro section.

My notes about the main text:
• L22, L41: non Biotic → abiotic
• References list has missing idents
• Species names should have italic font (L56, L138, L184, L324 and so on)
• L135: Seed → seed
• Plots at Figure 5 requires units (5A, 5D)
• L273: “LD50” missing subscript

Reviewer 3 ·

Basic reporting

Firstly, I appreciate the authors' effort in extensively revising the entire manuscript. The quality of the manuscript has obviously improved significantly. All of my comments have been fully addressed. However, there are still some minor problems with the formatting. I have provided some examples. Please note that the problems I have highlighted are not limited to the lines I have provided. Please revise these types of problems throughout the entire manuscript. After minor revisions and proofreading, the manuscript will be ready for publication.
1. L22: ‘abiotic’ instead of ‘non-Biotic’, same in line 41, please check the whole manuscript.
2. Spaces were not rightly added to the whole manuscript. For example, L49 & L53, there should be a space before the bracket. L39, the correct format is ‘(Li et al., 2021; Luo et al., 2019)’ Please follow this format, double check the whole manuscript.
3. L56: ‘E. nutans’ should be italicized.
4. Figure 1: Please adjust the location of the letters above the bar, make sure they are not overlapping.
5. In PDF, the figures were not clear, hopefully they will be clear enough for audiences when they were published.

Experimental design

no comment

Validity of the findings

no comment

---

## Round 0.3 · Minor Revisions

The authors should be improved the manuscript as per the reviewer's suggestion.

Reviewer 1 ·

Basic reporting

For specific suggestions, please refer to annotated PDF manuscript.

Experimental design

No comments

Validity of the findings

No comments

Annotated reviews are not available for download in order to protect the identity of reviewers who chose to remain anonymous.

·

Basic reporting

pass

Experimental design

Dots’ colors at PCA diagram (Figure 5) should be revised: blue for -0.49 and purple for -1.02 are almost the same.

Sections 4.3 and 4.4 still have vary similar title. Maybe these sections can be merged?

Refer to the raw data in Supplementary (spreadsheets files) in the manuscript. Attached XLSX files of figures data won't be available to the readers. Actually these data should be presented as tables (in Supplementary).

Validity of the findings

L451-452: please avoid repetition in Conclusion.
L68: seeds is → seeds are

Reviewer 3 ·

Basic reporting

I appreciate the authors' careful revision. My comments have been fully addressed, and I have no further comments. I recommend that this manuscript be accepted for publication.

Experimental design

no comment

Validity of the findings

no comment

---

## Round 0.4 · accepted · Accept

Overall, the manuscript is good and interesting for other researchers. The manuscript is acceptable for possible publication.